# How Can Operational Research Help to Eliminate Tuberculosis in the Asia Pacific Region?

**DOI:** 10.3390/tropicalmed4010047

**Published:** 2019-03-15

**Authors:** Anthony D. Harries, Ajay M. V. Kumar, Srinath Satyanarayana, Pruthu Thekkur, Yan Lin, Riitta A. Dlodlo, Rony Zachariah

**Affiliations:** 1International Union Against Tuberculosis and Lung Disease, 68 Boulevard Saint Michel, 75006 Paris, France; akumar@theunion.org (A.M.V.K.); SSrinath@theunion.org (S.S.); Pruthu.TK@theunion.org (P.T.); ylin@theunion.org (Y.L.); rdlodlo@theunion.org (R.A.D.); 2London School of Hygiene and Tropical Medicine, Keppel Street, London WC1E 7HT, UK; 3International Union Against Tuberculosis and Lung Disease, South-East Asia Office, C-6 Qutub Institutional Area, New Delhi 110016, India; 4Yenepoya Medical College, Yenepoya (Deemed to be University), University Road, Deralakatte, Mangalore 575018, India; 5International Union Against Tuberculosis and Lung Disease, No.1 Xindong Road, Beijing 100600, China; 6Special Programme for Research and Training in Tropical Disease (TDR), World Health Organization, Avenue Appia 20, 1211 Geneva 27, Switzerland; zachariahr@who.int

**Keywords:** operational research, tuberculosis, Asia Pacific, diagnosis, treatment, prevention, SORT IT

## Abstract

Broad multi-sectoral action is required to end the tuberculosis (TB) epidemic by 2030 and this includes National TB Programmes (NTPs) fully delivering on quality-assured diagnostic, treatment and preventive services. Large implementation gaps currently exist in the delivery of these services, which can be addressed and closed through the discipline of operational research. This paper outlines the TB disease burden and disease-control programme implementation gaps in the Asia-Pacific region; discusses the key priority areas in diagnosis, treatment and prevention where operational research can be used to make a difference; and finally provides guidance about how best to embed operational research within a TB programme setting. Achieving internationally agreed milestones and targets for case finding and treatment requires the NTP to be streamlined and efficient in the delivery of its services, and operational research provides the necessary evidence-based knowledge and support to allow this to happen.

## 1. Introduction

Despite progress over several decades in the global effort to control tuberculosis (TB), the disease continues to be a high-priority public health challenge, especially in low- and middle-income countries (LMIC). In 2017, an estimated 10 million people globally developed TB and 1.6 million TB patients died, making the disease one of the top 10 causes of death worldwide and the leading cause from any single infectious agent [1]. The international community, through the World Health Organization (WHO) End TB Strategy [2] and the United Nations (UN) Sustainable Development Goals (SDGs) [3], has pledged to end the Global TB Epidemic. Targets and interim milestones for meeting this ambitious goal include a 90% reduction in TB deaths and an 80% reduction in TB incidence by 2030 compared with 2015, and during this time, no TB-affected families should face catastrophic financial costs. While the global TB disease burden is decreasing, it is too slow and the world is currently not on track to meet the agreed milestones and targets [1].

What needs to be done? It seems clear that multi-sectoral action will be needed to end the global TB epidemic, that includes accelerating socio-economic development, developing and deploying a new TB vaccine, producing and using novel diagnostics and medicines for treatment, and expanding universal health coverage and universal access to health care [4]. Within this framework, National TB Programmes (NTPs) play an essential role, for they are responsible for delivering high quality and effective diagnostic, treatment and preventive services, which should interrupt the transmission of *Mycobacterium tuberculosis (MTB)* and slow down the incidence of TB.

To this end, The Stop TB Partnership in 2015 focused on measurable targets for NTPs, launching the 90-(90)-90 TB diagnostic and treatment targets as part of a Global Plan to accelerate the ending of the TB epidemic (Table 1) [5]. The UN High-Level Meeting (HLM) on the fight against TB in September 2018 produced a political declaration on TB with world leaders also committing to a number of actions by 2022 that include (i) successfully treating 40 million people for TB, including 3. 5 million children and 1.5 million persons with drug-resistant TB and (ii) providing TB preventive therapy to 30 million people, including 4 million children under the age of 5 years [6].

Health delivery services and TB control programme activities need to have country-wide coverage and to function at a high standard if these laudable targets are to be achieved. Unfortunately, health service coverage often fails to reach vulnerable and at risk-groups and many gaps exist in the implementation of services. Operational research is one way to address these challenges and assist in closing the gaps. It can be defined as research into interventions, strategies, and tools or new knowledge to enhance the quality, coverage and effectiveness of disease control programmes, health services or health systems in which the research is conducted [7]. In 2011, WHO published its “Priorities in Operational Research to improve Tuberculosis care and control” to serve as a guiding framework for NTPs [8], and there are several examples from Africa and Asia of how operational research conducted within TB control programmes can lead to improved programmatic performance with increases in case finding and successful treatment outcomes [9,10,11,12]. Specific examples of operational research that led to national policy change include: (i) HIV testing and cotrimoxazole preventive therapy in Malawi which was associated with significantly reduced country-wide mortality in HIV-infected TB patients [13]; (ii) HIV testing in patients with presumptive TB in India which led to national implementation of testing and the potential for earlier referral for antiretroviral therapy in this group of patients [14,15]; and (iii) screening of TB patients for diabetes mellitus in India which led to widespread implementation of screening for diabetes in certain regions of the country [16].

The objectives of this paper are to (i) outline the TB disease burden and disease-control programme implementation gaps in the Asia-Pacific Region, (ii) discuss the priority areas in which operational research can make a difference and (iii) provide guidance about how to get research embedded within a programme setting and used for changing policy and practice.

## 2. TB Burden and Disease Control Efforts in the Asia-Pacific Region

The 2018 WHO Global TB Report provides all the relevant data for the populations, the burden of TB, and progress with TB control efforts in Asia-Pacific countries [1]. The Asia-Pacific comprises both the WHO South-East Asia and the Western Pacific Regions where 3.9 billion people (52% of the world’s population) live. In 2017, the estimated incidence of TB in the Asia-Pacific Region was 6.2 million (62% of the global total) with deaths from TB estimated at 760,000 (49% of the global total) [1]. Five countries in the Asia-Pacific Region (India, China, Indonesia, Philippines, and Pakistan) account for 55% of the estimated global TB burden. Most countries in the region have established NTPs but as discussed earlier there are important implementation gaps in service delivery.

In 2017, only 4.3 million new TB patients in the Asia-Pacific Region were notified, meaning that 1.9 million (31% of the estimated disease burden) were either not diagnosed or were diagnosed but not notified to national programmes [1]. India and Indonesia have the largest gaps globally between estimated and notified cases. There were 306,000 estimated people with MDR/RR-TB (Multidrug Resistant TB/Rifampicin Resistant-TB; namely, TB resistant to both isoniazid and rifampicin or rifampicin alone) of which 76,487 (25%) were notified—of these, 2886 were diagnosed with XDR-TB (Extensively drug resistant TB; namely, MDR-TB with additional resistance to second-line drugs, such as fluoroquinolones and injectable second-line agents). There were 183,000 estimated patients with HIV-TB (human immunodeficiency virus-TB) coinfection, of which 74,004 (40%) were notified.

Not all those diagnosed and notified with TB are registered and enrolled for treatment, and treatment success rates of those who are registered fall below what is expected according to global targets (see Table 2) [1]. The main adverse outcomes contributing to these low treatment success rates include death, being lost to follow-up, not being evaluated at the end of treatment, or no data being reported [1].

TB/HIV collaborative activities in 2017 included only 50% of notified TB patients being HIV-tested (3.5% were HIV-positive), while 72% of notified HIV-positive TB patients were started on antiretroviral therapy (ART). Of people living with HIV and newly enrolled in HIV care, 12% and 38% were provided with TB preventive treatment in the South-East Asia and Western Pacific Regions, respectively [1]. Finally, 2018 was the first year in which WHO collected data on the coverage of TB preventive treatment amongst eligible household contacts under 5 years of age. The data was patchy from the Asia Pacific Region, but India, Thailand, and Myanmar had estimated coverage rates of 11%, 5% and 2% respectively [1].

## 3. Operational Research to Improve TB Programme Performance

There are several key areas that should be prioritised for operational research within NTPs in the Asia-Pacific Region and these relate to important and essential deliverables that all NTPs should aspire to in order to meet WHO and SDG targets and end TB [17].

### 3.1. Finding Persons with Presumptive TB and Making a Rapid Diagnosis

Finding more people with TB depends on (i) identifying and streamlining the pathways that persons with presumptive TB take to access health services; (ii) improving the screening of persons with presumptive TB; (iii) introducing new tools and diagnostic algorithms to increase the accuracy and rapidity of diagnosis; and (iv) engaging in cost-effective active case finding in high-risk and vulnerable groups.

#### 3.1.1. Pathways to Care and Screening 

Operational research should be used to assess the pathways that patients take to access care, including the use of the allopathic and traditional sector. More information is needed about the extent and time spent with traditional healers, who then need to be engaged and encouraged to collaborate and participate in shortening the time taken from symptom onset to diagnosis. NTPs detect most of their TB patients through a strategy of passive case finding—namely waiting for symptomatic patients to seek health care at health facilities and instituting well established guidelines for screening and investigation. Recent assessments of this process within the health sector show that it performs badly, with health care staff often failing to screen those who should be screened and not properly investigating those identified with presumptive TB [18]. Reducing this “pre-diagnostic loss to follow-up” is a fertile area for operational research.

#### 3.1.2. TB Diagnostic Tools 

Sputum smear microscopy has been the mainstay of TB diagnosis for over 100 years, but it is cumbersome for technicians and patients, relatively less sensitive, and unable to diagnose drug-resistant TB. Molecular technology is fast taking over, especially the use of the fully-automated and commercially available cartridge-based nucleic acid amplification test called Xpert MTB/RIF (Cepheid Inc, Sunnyvale, CA, USA) which allows a confirmed diagnosis of TB within 2 h along with information about the presence or absence of rifampicin resistance (used as a proxy for diagnosing MDR-TB) [19]. WHO now recommends that Xpert MTB/RIF be used as the initial diagnostic test for all people requiring investigation for TB [20]. Newer molecular diagnostic platforms and instruments are being developed and deployed that are more sensitive: These include Xpert MTB/RIF Ultra [21], portable and battery-operated instruments (Xpert OMNI) that would potentially allow decentralization of diagnosis to the primary health care level and community [22], and a new assay capable of diagnosing *Mycobacterium tuberculosis* mutations associated with resistance to isoniazid, fluoroquinolones, and aminoglycosides [23].

Despite the rapid scale up of Xpert MTB/RIF globally in the last few years [1], it is unlikely that this will completely replace smear microscopy for the foreseeable future, and in most NTPs, the two diagnostic methods will co-exist. Operational research needs to be undertaken to work out how the two methods, including newer molecular platforms, should best complement each other so as to provide a comprehensive, accessible, and continuous service for the maximum number of patients [24]. Other operational research questions include: how to ensure health care workers have and maintain a high clinical suspicion of TB; how to collect good quality sputum specimens and ensure efficient transportation to diagnostic centres; how to make the most efficient use of the molecular technology assays and instruments so that they deliver at maximum potential for adult and childhood diagnosis; what is the role of national reference laboratories for mycobacterial culture and drug susceptibility testing and line probe assays; and how should chest radiography best be used [25].

#### 3.1.3. Active Case Finding (ACF)

This is an approach which proactively seeks to find and diagnose persons with TB, and WHO has provided guidelines for the systematic and active screening of high-risk groups and household contact investigations (Table 3) [26,27]. While NTPs will have to decide what they can do in this regard based on their human and financial resources and capabilities, people living with HIV (PLHIV) and household contacts should be prioritised. In 2017, only one in four household contacts under 5 years of age was screened for TB and only 51% of PLHIV were actually notified with TB against the number of incident TB patients estimated among PLHIV [1,28,29]. These figures suggest enormous room for improvement.

The 2016 WHO Guidelines on the use of antiretroviral (ARV) drugs for treating and preventing HIV infection recommend that all PLHIV start antiretroviral therapy (ART) regardless of clinical stage or CD4 cell count [30]. Thus, the screening for TB in PLHIV will increasingly take place in the context of ART. In busy ART clinics, TB screening and subsequent investigations may end up being done hurriedly and poorly, and operational research is needed to determine feasible and effective ways for over-worked staff to do this task better. In PLHIV with more advanced immunodeficiency who may be hospitalised, the diagnosis of TB is difficult with autopsy studies showing that a large proportion of TB is missed in this group [31]. In this regard, the measurement of urine lipoarabinomannan (LAM), one of the cell wall lipopolysaccharide components of *Mycobacterium tuberculosis*, is a promising bed-side test in PLHIV. The measurement can be done with a Determine TB-LAM test strip (Alere, Waltham, MA, USA), and this produces results in 30 min with high specificity and increasing sensitivity as the CD4 cell counts decrease to below 100 cells/µL [32]. Bedside urine LAM on its own has been found useful to guide anti-TB treatment in hospitalised PLHIV and reduce early mortality [33]. Urine LAM combined with urine Xpert MTB/RIF has also resulted in an increase in TB diagnosis and treatment and a reduced overall mortality in key subgroups of PLHIV [34]. PLHIV with advanced disease are often so sick that they cannot cough up sputum, and the use of urine as an easy to collect specimen is a novel step forward. Operational research is needed to assess the best ways of integrating these new diagnostic approaches within the health system.

NTPs need to step up their screening of household contacts of index bacteriologically-confirmed TB patients, focusing on children < 5 years of age, and work out how best to do this based on the local resource-constrained context. A recent systematic review found that all household contacts (including adults) are at high risk of developing TB [35], and a cluster randomised trial in Vietnam showed the benefit and positive yield of repeated household visits at which household members were assessed by symptoms, physical examination and chest radiography [36]. NTPs must seriously consider implementing and monitoring this ACF amongst household contacts of index TB cases.

### 3.2. Providing Rapid and Effective Treatment to Those Diagnosed with TB

Providing rapid, comprehensive and effective treatment depends on (i) efficient and timely linkages between diagnostic and treatment services; (ii) short, simple, patient-friendly and effective regimens that have been formally assessed in clinical trials and observational studies; (iii) accounting for important co-morbidities, such as HIV and diabetes mellitus (DM); and (iv) eliminating adverse programmatic outcomes, such as lost to follow-up and patients “not evaluated”. An important programmatic step is to ensure that the monitoring of treatment outcomes uses diagnosed patients as the denominator rather than those registered for treatment, as this is the only way to ensure progress against the 90-(90)-90 Stop TB Partnership targets [5].

#### 3.2.1. Pre-treatment Loss to Follow-up 

Globally, between 4% and 38% of patients with laboratory-detected sputum smear-positive or culture-positive TB fail to start treatment, and this is termed pre-treatment loss to follow-up [37]. For those who do get treated, the time between confirmed diagnosis and treatment initiation can also be lengthy, and this compromises patient care and increases the risk of *Mycobacterium tuberculosis* transmission within families and communities. NTPs need to engage in operational research to identify the extent and reasons for pre-treatment loss to follow-up, put in place practical solutions to minimise the losses, and reassess whether these solution have indeed made a difference.

#### 3.2.2. Short, Simple and Patient-Friendly Treatment 

New patients with drug-susceptible TB are treated with a standardised regimen of isoniazid and rifampicin for 6 months, supplemented with pyrazinamide and ethambutol for the first 2 months [38]. Attempts to shorten the treatment to 4 months by using fluoroquinolones and rifapentine have been unsuccessful [39,40,41]. The drugs in the 6-month regimen (with dosages standardised by body weight and in fixed-dose combinations) are given under direct observation (DOT) usually in the community, and how this is best done can be facilitated through operational research.

WHO has recently published updated guidance on the treatment of RR/MDR-TB [42]. The traditional longer regimens are usually for 18–24 months, but shorter regimens of 9–12 months are now recommended, provided resistance to fluoroquinolones and second-line injectable agents has been excluded or is considered highly unlikely. Well conducted operational research has been valuable to date [43,44] and will be needed in the future to determine (i) what are the most suitable short regimens for local context; (ii) how best to replace injectable agents with oral drugs, such as bedaquiline, delamanid and linezolid; and (iii) how in resource-limited settings, NTPs can implement and read the electrocardiograms deemed necessary to monitor prolonged QT intervals that may occur with high dose moxifloxacin, an essential drug in the shorter MDR-TB regimens [45].

#### 3.2.3. Co-Morbidities

Two important co-morbidities that affect TB treatment outcomes are HIV and DM [46]. Both these co-morbid diseases increase case fatality and the risk of recurrent disease in patients with TB and pose challenges for joint treatment in terms of drug–drug interactions and adverse events. For HIV/AIDS, the key interventions are HIV testing for all presumptive and diagnosed TB patients, timely initiation of ART and administration of cotrimoxazole preventive therapy (CPT) which further augments the effects of ART. More needs to be learnt about the interactions between anti-TB and ARV drugs, especially between rifampicin and new integrase inhibitors, such as dolutegravir which is increasingly being promoted as a component of first-line ART [47]. The optimal ways to improve treatment outcomes in patients with DM-associated TB need further research. Limited evidence to date suggests that better management of DM and use of metformin as the oral anti-diabetes medication reduces the risk of death during TB treatment [48,49]. High quality health service-related operational research is required to understand how to better integrate and decentralise HIV, DM and TB services, aiming if possible for a “one stop shop service” that serves the needs of patients with dual or triple infection. Onto this can be added ways to manage alcohol dependence and intravenous drug use, help patients quit tobacco smoking, and provide appropriate mental health care.

#### 3.2.4. Eliminating Adverse Programmatic Outcomes

New definitions for treatment outcomes were published by the WHO in 2013 (Table 4) [50]. In the cohorts evaluated in that year, 5% and 2% of patients in the South-East Asia Region were lost to follow-up and not evaluated respectively with figures from the Western Pacific being 2% and 4% respectively [51]. Prospective operational research to determine what has happened to patients recorded as lost to follow-up can provide the NTP with important information and potential solutions to reduce this particular outcome for drug-susceptible and drug-resistant disease [52,53]. “Not evaluated” is an outcome given when no treatment outcome is assigned and includes patients who are transferred out to another treatment unit and whose treatment outcome is unknown. Transfers during anti-TB treatment are common and are often associated with no treatment outcome in the TB treatment register [54,55]. Operational research can assess the use of mobile phones and other modern communication technology in reducing this correctable adverse outcome and improving treatment success rates.

### 3.3. Preventing TB

Preventing TB involves (i) implementing effective infection control practices, (ii) treating latent TB infection and iii) addressing factors and diseases that increase the risk of developing TB.

#### 3.3.1. Infection Control 

In health care facilities and other congregate settings, such as prisons, there can be intense transmission of *Mycobacterium tuberculosis* and properly implemented infection control measures can prevent or at least limit the amount of exposure to *Mycobacterium tuberculosis* amongst non-infected individuals. The WHO has published TB infection control guidelines which focus on early identification, isolation and treatment of those with presumptive TB, health facility infrastructure modifications, better organization to reduce patient congestion and personal protective measures for health workers [56]. In general, these measures are poorly applied and operational research should be used to ensure much better implementation along with regular monitoring of the actions and activities. This includes adhering to an important health facility monitoring indicator which is the number and proportion of health care staff who develop TB each year.

#### 3.3.2. Treating Latent TB Infection 

Latent TB infection (LTBI) is defined as a state of persistent immune response to stimulation by *Mycobacterium tuberculosis* antigens with no evidence of clinically manifest active TB [57]. The diagnosis of LTBI is made on the basis of a positive tuberculin skin test or positive interferon-gamma-release assay, both of which are challenging to use in resource-limited settings [58]. Once diagnosed, the treatment of LTBI reduces the risk of progression to active disease. WHO has recently released updated guidance on the programmatic management of LTBI with the high-risk groups recommended for treatment—PLHIV and household contacts of TB patients—being the same as those recommended for active case finding. Isoniazid monotherapy for 6 months is the most commonly used treatment although shorter regimens of 3 months with rifampicin and isoniazid or rifapentine and isoniazid are also proposed [59]. The details of how best to educate persons about latent TB treatment, follow-up on treatment adherence and monitor and manage any adverse events, especially drug-induced hepatitis, require operational research.

#### 3.3.3. Addressing Other Diseases/Socio-Economic Determinants that Increase the Risk of TB 

This involves close examination of important determinants of TB such as HIV, DM, smoking, alcohol dependence, and malnutrition. The classic example is the use of ART in PLHIV which reverses the immune dysfunction associated with HIV and, as a result, has a potent TB preventive effect [60]. It is now clear from randomised controlled trials that the effects of ART in reducing TB incidence may be augmented with the addition of isoniazid preventive therapy [61,62]. There are many pertinent operational research questions to be answered here that range from the optimal public health approach to scaling up ART to when to start and then monitor the addition of isoniazid preventive therapy.

### 3.4. The Private-for-Profit Sector

Many people with presumptive TB seek and receive care from private providers that can be both formal and informal, and this is especially the case in the Asia-Pacific region [63]. As a result, the WHO has developed a global strategy called Public-Private Mix (PPM) which aims to mix “clinical tasks” (the referral of symptomatic patients, diagnosis of TB and the prescribing of treatment) and “public health tasks” (quality assurance, recording and reporting) and link private sector providers to local public sector TB programmes [64]. Significant scale up, however, has yet to happen. Updated WHO guidance on national action plans was released in 2017 [65], and operational research can help to turn the recommendations into ground reality.

## 4. Embedding Operational Research within the NTP

Embedding operational research within the NTP is the best way to ensure that local and relevant research is implemented according to the country’s needs and priorities [66]. Malawi, Benin, Vietnam and more recently Pakistan and India are good examples of how this has worked in practice [9,10,11,12,67]. There are some key steps that need to be addressed if such an initiative is to be successful, and these are shown in Table 5.

### 4.1. Setting up the Operational Research Programme 

The initial series of steps involves obtaining political commitment for the initiative, integrating operational research into the national TB strategic plan, and ensuring that there is a dedicated budget line. Without a budget specifically earmarked for operational research, it is unlikely that any significant research activity will take off. Funds are needed for human resources, the office with computer and internet, transport, meetings to discuss and prioritise the necessary research and design of the appropriate studies, ethics protocol reviews, the conduct of the operational research, open access publications, dissemination meetings, skills training, and attendance at national and international conferences. A sound ethics framework must underpin the operational research programme, with an understanding that interviews with patients require informed consent and use of routinely collected secondary data needs ethical review and approval to ensure the confidentiality of data [68,69]. Developing a good relationship with National Science and Ethics Committees is essential in this regard so that the use of routinely collected secondary data that provides good programmatic information can be fast tracked for ethics approval. Unfortunately, in many resource-poor countries, the local capacity for ethics committee review and oversight is underdeveloped. This is particularly the case in a number of Pacific Island nations where ethics committees have not been established [70]. This is a challenge for the optimal development of culturally appropriate and relevant research and the building up of local ethics committees should be an integral part of research development.

### 4.2. Appointing Skilled Research Officers and Capacity Building

The next most important step is to appoint a skilled operational research officer to lead and coordinate the research endeavour. As described earlier, the appointment of one operational research officer within the NTP in Benin and Vietnam was successful in generating national, good quality, relevant research [10,11], but building a critical mass of two or more research officers within an institution can help to foster an institutional culture of research and result in significant scale up of research projects and publications [71].

Operational research methodology can vary from simple cross-sectional, cohort or case control studies to slightly more complicated “before and after” studies or longitudinal assessments. More sophistication can be applied with pragmatic cluster randomised trials, stepped wedge designs and implementation frameworks such as the RE-AIM (Reach, Effectiveness, Adoption, Implementation and Maintenance) model [72], but these will almost certainly require partnerships and assistance from academic and research institutions.

Building capacity for programmatic research officers to undertake and publish simple cross-sectional, cohort or case control studies using routinely collected data, and getting them rapidly up to speed is essential. The SORT IT (Structured Operational Research and Training Initiative) model is one way of doing this. In brief, the model hinges on SORT IT courses which consist of three 1-week modules over 9 months, with well-defined milestones and targets, and strong hands-on mentorship enabling participants to develop ethically-approved research protocols that are then taken through to completion, publication and dissemination to a national/international audience [73]. Participants are followed-up for 18–24 months after each course to determine whether their projects have impacted on policy and practice and whether they are still continuing to do operational research: In the latest assessment, 45% of SORT IT alumni had completed a new research project and 36% had published a new scientific paper within 18 months of course completion [74]. SORT IT courses started in 2009 and as of 31 December 2018, there had been 69 courses (some of which are still on-going) which had enrolled 746 participants from 90 countries around the world. Of these participants, 383 were from 30 countries in the Asia Pacific Region. Table 6 shows the number of TB operational research projects in this region which were developed by the end of 2018, with outputs and outcomes. Support for SORT IT courses at the national level would be a cost-effective way of rapidly building a sustainable research capacity while at the same time fostering the implementation of much needed operational research. For ongoing guidance with developing and implementing research studies, research officers can also make use of the WHO-TDR guidelines on operational research and the WHO-TDR implementation research toolkit which can be downloaded from the internet and used to help with methodology [75,76].

### 4.3. Management and Monitoring Structures 

A formal management and monitoring structure allows key research questions to be developed and provides a framework to monitor what is actually happening. In Malawi, a TB Programme Management Group was set up that met every 4 weeks with a large part of the agenda focused on operational research [9]. Chaired by the NTP, national stakeholders (that included representatives from other disease programmes such as HIV/AIDS, academia and non-governmental organizations) were invited as sitting members and international institutions such as the WHO and European Schools of Tropical Medicine were invited to attend if they were in the country. Decisions were made about what research was needed for the programme and who would conduct the research, and the Management Group monitored the progress made. There was a culture of moving fast on completing and writing up research findings and making decisions about impact on policy and practice. There was an annual report on operational research which followed a standardised template as shown in Table 7.

## 5. Conclusions

NTPs play a crucial role in ending the TB epidemic. Basic TB control principles need to be adhered to, not just in planning, but also in service implementation. The large gaps in diagnostic, treatment and preventive services in the public and private sectors must be quantified and the reasons identified. Quality of care often leaves much to be desired, and NTPs need to measure this dimension of health care and invest in quality assurance [77]. Operational research embedded within the TB programme offers the best way of achieving this. For every challenge or constraint that is identified, a feasible solution must be found and translated into policy and practice. The cycle of operational research then continues, assessing whether the new policy or practice has made a difference to programme performance and health-related outcomes. The new diagnostic and treatment tools that will be developed need to be rapidly deployed in the field if international targets are to be reached in time. Operational research will be required to determine the best ways to integrate new tools within programme activities and monitor their effectiveness and thereby ensure that they can be scaled up.

## Figures and Tables

**Table 1 tropicalmed-04-00047-t001:** The 90-(90)-90 targets for Global Tuberculosis Control.

Reach, diagnose and treat at least 90% of all people with tuberculosis ^a^ As a part of this approach, reach, diagnose and treat at least (90%) of the key populations ^b^ Achieve at least 90% treatment success for all people diagnosed with tuberculosis ^c^

^a^—includes people with both drug-susceptible and drug-resistant tuberculosis as well as people who require preventive therapy; ^b^—includes vulnerable, underserved and at-risk populations which vary depending on the country context; ^c^—includes achieving 90% treatment success among people diagnosed with both drug-susceptible and drug-resistant TB as well as people who require TB preventive therapy. From the Stop TB Partnership and adapted from Reference [5].

**Table 2 tropicalmed-04-00047-t002:** Treatment success rates for patients with tuberculosis, including those with HIV-positive and drug-resistant disease, in the Asia-Pacific region.

TB Case Type and Cohort	Number in the Cohort	Treatment Success (%)
New and relapse TB registered in 2016	3,966,632	(81)
HIV-positive TB registered in 2016	70,867	(70)
MDR/RR-TB registered in 2015	45,708	(51)
XDR-TB registered in 2015	2256	(29)

Adapted from Reference [1].

**Table 3 tropicalmed-04-00047-t003:** Active tuberculosis case finding for high-risk or vulnerable groups.

**High priority for active case finding**
Household and other close contacts.
People living with HIV at each visit to a health facility.
Current and past workers who have been exposed to silica in their workplaces.
**Active case finding if resources permit**
Prisoners and people in other penitentiary institutions People who have untreated fibrotic lesions on chest radiography.
People who are seeking health care or who are already in health care in high TB prevalent countries and who belong to selected risk groups: these include malnutrition, diabetes mellitus, alcohol dependence, tobacco smoking, intravenous drug use, chronic renal disease, a previous history of tuberculosis and old age
Others - people living in urban slums, homeless people, people living in remote areas, some indigenous populations, and migrants/refugees.

Adapted from Reference [27].

**Table 4 tropicalmed-04-00047-t004:** Revised definitions of treatment outcomes for patients with drug susceptible tuberculosis: 2013 WHO Guidance.

Treatment Outcome	Revised Definition
Cure	A bacteriologically confirmed tuberculosis patient at the beginning of treatment who was found to be smear- or culture-negative in the last month of treatment and on at least one previous occasion.
Treatment completed	A tuberculosis patient who completed treatment without evidence of failure BUT with no record to show that sputum smear or culture results in the last month of treatment and on at least one previous occasion were negative, either because the tests were not done or because the results are unavailable.
Treatment failure	A tuberculosis patient whose sputum smear or culture was positive at month 5 or later during treatment.
Died	A diagnosed tuberculosis patient who died for any reason before starting or during the course of treatment.
Lost to follow up	A diagnosed tuberculosis patient who did not start treatment or whose treatment was interrupted for 2 consecutive months or more.
Not evaluated	A tuberculosis patient for whom no treatment outcome was assigned. This includes patients “transferred-out” to another treatment unit as well as patients for whom the treatment outcome was unknown to the reporting unit.
Treatment success	The sum of *cured* and *treatment completed*.

Adapted from Reference [50].

**Table 5 tropicalmed-04-00047-t005:** Key steps for embedding operational research in a National TB Programme.

Political commitment for embedding and learning from operational research in the Programme.
Integration of operational research into the National TB strategic plan.
Dedicated budget line for the operational research unit.
Identify resources for conducting and disseminating research and for training.
Good relationship with National Ethics Board to fast track use of secondary data.
Appointment of skilled and dedicated research officer(s) to lead and coordinate the research agenda.
Capacity building opportunities and mentorship available for research officers.
Management and monitoring structure that includes national /international institutions.
Research questions address constraints to TB care and prevention and are planned within the NTP.
Encourage a culture of moving fast and making decisions on papers, policy and practice.
Regular evaluation and reporting on research outputs, outcomes and impact.

**Table 6 tropicalmed-04-00047-t006:** Tuberculosis research projects generated through SORT IT courses in the Asia Pacific Region: 2009–2018.

Characteristics	Number	(%)
Tuberculosis projects undertaken	177	
Projects completed and manuscripts submitted to journals ^a^	158	(89)
Papers published in peer-reviewed journals ^b^	117	(74)
Papers eligible for policy and practice assessment ^c^	111	
Papers assessed for policy and practice impact ^d^	101	(91)
Papers judged to have had an impact on policy and practice ^e^	71	(70)

^a^ percentage of tuberculosis projects undertaken that were submitted to journals; ^b^ percentage of papers submitted to journals that were published (this is a cross-sectional analysis and several papers are still under peer review); ^c^ number of submitted papers eligible for policy/practice assessment (this takes place 18 months after submission to a journal); ^d^ percentage of eligible papers that were assessed for policy and practice; ^e^ percentage of assessed papers judged to have an impact on policy and practice (judgement based on self-assessment and follow-up calls from the Centre for Operational Research at the Union).

**Table 7 tropicalmed-04-00047-t007:** Proposed framework for the regular evaluation of operational research within a National TB Programme.

Research protocols developed and approved by National Ethics Committees						
↘	Research studies completed					
	↘	Research papers submitted to peer-reviewed journals				
		↘	Research papers published			
			↘	Research findings disseminated		
				↘	Changes in policy and practice	
					↘	Evaluation of effect on programme performance.

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
