# Peer review of "How Can Operational Research Help to Eliminate Tuberculosis in the Asia Pacific Region?"

_tropicalmed, 2019, doi:10.3390/tropicalmed4010047_

Round 1

Reviewer 1 Report

The manuscript provides a thorough and comprehensive approach to identifying the root causes and gaps in implementation of policies outlined in NTPs. It also focuses on specific areas in which operational research can have the most impact. The manuscript is well researched and the roadmap that is provides is timely in terms of meeting the targets set out by NTPs.

Author Response

We thank the reviewer for the positive comments. We note that there are no revisions to be made. On behalf of the co-authorship, best wishes Tony Harries 

Reviewer 2 Report

This is a helpful overall review for this special TB edition of the journal. It does not present any primary data, and is not especially original, but does provide a focus on appropriate operational research areas on which to focus in the Asia Pacific region.

No methods are provided. Therefore the sources of data e.g. in section 2, are not entirely unclear. Also in lines 101-107, it is unclear which Asia-Pacific countries the data relate to.  

in lines 214-215, linezolid can also be added – see https://www.ncbi.nlm.nih.gov/pubmed/30477961

it would have been helpful to include more on actual operational research models. Identifying priority research areas is the easy part. Designing a study fit for purpose is the hard part. When clinicians or clinician-researchers in these settings wish to undertake operational research in any of the priority areas indicated in this paper, what sort of frameworks can they use? Where can they seek guidance on methodological approaches suited to their research questions? Not everyone has access to SORT-IT training. Examples of evaluation frameworks or implementation research models could be helpful. e.g. a table describing some different approaches (before-and after studies, longitudinal analyses, cluster randomised trials including stepped wedge designs; implementation frameworks such as the Consolidated Framework for Implementation Research or the RE-AIM model), with examples of TB studies using that approach cited.  

Author Response

Response to Reviewer 2

This is a helpful overall review for this special TB edition of the journal. It does not present any primary data, and is not especially original, but does provide a focus on appropriate operational research areas on which to focus in the Asia Pacific region.

Response:                                                                                                                                        

Thank you for the positive comments. This was not meant to be an original paper but a perspective paper outlining the importance of operational research to eliminate TB in the Asia Pacific.

No methods are provided. Therefore the sources of data e.g. in section 2, are not entirely unclear. Also in lines 101-107, it is unclear which Asia-Pacific countries the data relate to.

Response:                                                                                                                                                 

This is not an original research paper and therefore a methods section was not provided. However, this point is very valid. We have now started Section 2 with a new sentence as follows in lines 93-94: “The 2018 WHO Global TB Report provides all the relevant data for the populations, the burden of TB and progress with TB control efforts in the Asia-Pacific countries” and we have referenced this with the Global TB Report [1]. As explained in this sentence and in the sentence below, the Asia Pacific includes the WHO South East Asia and Western Pacific regions and the countries are listed in the WHO Report.

In lines 214-215, linezolid can also be added – see https://www.ncbi.nlm.nih.gov/pubmed/30477961

Response:                                                                                                                                                            In the current version of the paper which we have received from the journal, lines 214-215 refer to standardized regimens for drug-susceptible TB. We think the reviewer is referring to the statement on lines 226-227 about second-line drugs such as bedaquiline and delaminid. We have therefore added “linezolid” here.

It would have been helpful to include more on actual operational research models. Identifying priority research areas is the easy part. Designing a study fit for purpose is the hard part. When clinicians or clinician-researchers in these settings wish to undertake operational research in any of the priority areas indicated in this paper, what sort of frameworks can they use? Where can they seek guidance on methodological approaches suited to their research questions? Not everyone has access to SORT-IT training. Examples of evaluation frameworks or implementation research models could be helpful. e.g. a table describing some different approaches (before-and after studies, longitudinal analyses, cluster randomised trials including stepped wedge designs; implementation frameworks such as the Consolidated Framework for Implementation Research or the RE-AIM model), with examples of TB studies using that approach cited.  

Response:                                                                                                                                                              Thank you. We have tried to strengthen this section with additional narrative and references which have also been suggested by the second reviewer. In terms of evaluation frameworks, we have also added narrative in lines 341-346 as follows: “Operational research methodology can vary from simple cross-sectional, cohort or case control studies to slightly more complicated “before and after” studies or longitudinal assessments. More sophistication can be applied with pragmatic cluster randomised trials, stepped wedge designs and implementation frameworks such as the RE-AIM (Reach, Effectiveness, Adoption, Implementation and Maintenance) model [ref], but these will almost certainly require partnerships and assistance from academic and research institutions.” We feel that the paper is already long at over 4500 words with seven tables and would prefer not to go into this particular aspect in any more detail or add in more tables. We hope the reviewer and editor can accept our stance here.

Reviewer 3 Report

The sentence spanning lines 69-71 needs rewording. The link between the "solutions to TB epidemic" (lines 55-68) and the services being studied through operational research (lines 69-78) is not as smooth as it could be. Perhaps some examples of existing programs and services would be helpful as well as a link to how operational research could support a reworking of multi-sectoral action (referred to on line 55).

In general, the examples of instances where operational research would be of use are good; however, it would be useful to the user to have references to examples (not necessarily in TB) where operational research was done well or changed outcomes for the better, or examples of challenges to carrying out OR and how they can be overcome. References to guides for operational or implementation research would also be useful.

A discussion on designing operational research studies and obtaining ethical approval in the context of TB would could also be strengthened in section 4 on setting up the operational research programme. (Perhaps an exploration of common ethical concerns).

If there is room, the two pieces I would expand on are the cost efficacy of dedicated health budget lines for operational research and prioritisation approaches for operational research studies.

Overall, I think this is a useful contribution. I especially like table 5. Some examples where OR was done well and references for learning how to embed OR in programmes and troubleshoot as necessary would strengthen the impact of the piece. 

Author Response

Reviewer 3: Response to Comments

The sentence spanning lines 69-71 needs rewording. The link between the "solutions to TB epidemic" (lines 55-68) and the services being studied through operational research (lines 69-78) is not as smooth as it could be. Perhaps some examples of existing programs and services would be helpful as well as a link to how operational research could support a reworking of multi-sectoral action (referred to on line 55).

Response:                                                                                                                                                      Thank you. We have re-written this section to better link the statements about solutions to the TB epidemic and services being addressed through operational research. We have now stated starting on line 69: “Health delivery services and TB control programme activities need to have country-wide coverage and to function at a high standard if these laudable targets are to be achieved. Unfortunately, health service coverage often fails to reach vulnerable and at risk-groups and many gaps exist in the implementation of services. Operational research is one way to address these challenges and assist in closing the gaps.” We have already provided a short statement with four good references about how operational research can support existing programs and services (lines 77-81), but we have expanded on this to provide in the narrative text some concrete examples which we have supported with four additional references (lines 81-86). We are not sure about how operational research can support a reworking of multi-sectoral action and have therefore not tackled this query. However, this was not really the purpose of the paper which states in lines 58-61 that we are assessing the value of operational research within the context of National TB Program activities. 

In general, the examples of instances where operational research would be of use are good; however, it would be useful to the user to have references to examples (not necessarily in TB) where operational research was done well or changed outcomes for the better, or examples of challenges to carrying out OR and how they can be overcome. References to guides for operational or implementation research would also be useful.

Response:                                                                                                                                                       Thank you. As mentioned earlier we have provided some additional narrative with four additional references in lines 81-86 as follows:  “Specific examples of operational research that led to national policy change include:- i) HIV testing and cotrimoxazole preventive therapy in Malawi which was associated with significantly reduced country-wide mortality in HIV-infected TB patients [13]; ii) HIV testing in patients with presumptive TB in India which led to national implementation of testing and the potential for earlier referral for antiretroviral therapy in this group of patients [14,15]; and iii) screening of TB patients for diabetes mellitus in India which led to widespread implementation of diabetes screening in certain regions of the country [16].” Thank you also for the suggestion about referencing published guides for operational or implementation research! In the section on operational research capacity building after discussing the SORT IT model in lines 349-364, we have drawn attention to two useful guidelines from WHO/TDR in lines 364-367 and have provided the references for these in the Bibliography.

A discussion on designing operational research studies and obtaining ethical approval in the context of TB could also be strengthened in section 4 on setting up the operational research program (Perhaps an exploration of common ethical concerns).

Response:                                                                                                                                                      Thank you and excellent point. We have expanded on these ethics issues in lines 324-327 as follows: “A sound ethics framework must underpin the operational research program, with an understanding that interviews with patients require informed consent and use of routinely collected secondary data needs ethical review and approval to ensure confidentiality of data {68,69]” and in lines 330-334 – “Unfortunately, in many resource-poor countries the local capacity for ethics committee review and oversight is underdeveloped. This is particularly the case in a number of Pacific Island nations where ethics committees have not been established [70]. This is a challenge for the optimal development of culturally appropriate and relevant research and the building up of local ethics committees should be an integral part of research development.”  We have supported these two statements with three additional references.

If there is room, the two pieces I would expand on are the cost efficacy of dedicated health budget lines for operational research and prioritization approaches for operational research studies.

Response:                                                                                                                                                      Thank you. On lines 320-321, we have made a short but frank comment that without a dedicated budget line it is unlikely that any significant research activity will take off and in the next sentence have explained what the funds would be used for. We have added some short statements in the section on “embedding operational research into the NTP” about how to prioritize operational research studies.

I think this is a useful contribution. I especially like table 5. Some examples where OR was done well and references for learning how to embed OR in programs and troubleshoot as necessary would strengthen the impact of the piece. 

Response:                                                                                                                                                              Thank you. We hope that by addressing the points you have raised earlier that we have indeed strengthened the paper.